# Heterogeneous Strain Distribution in the Subchondral Bone of Human Osteoarthritic Femoral Heads, Measured with Digital Volume Correlation

**DOI:** 10.3390/ma13204619

**Published:** 2020-10-16

**Authors:** Melissa K. Ryan, Sara Oliviero, Maria Cristiana Costa, J. Mark Wilkinson, Enrico Dall’Ara

**Affiliations:** 1Department of Oncology and Metabolism, Mellanby Centre for Bone Research, University of Sheffield, Sheffield S10 2TN, UK; s.oliviero@sheffield.ac.uk (S.O.); mcfdc89@gmail.com (M.C.C.); j.m.wilkinson@sheffield.ac.uk (J.M.W.); e.dallara@sheffield.ac.uk (E.D.); 2INSIGNEO Institute for in silico Medicine, University of Sheffield, Sheffield S10 2TN, UK; 3Medical Device Research Institute, Flinders University, Adelaide 5042, Australia

**Keywords:** hip osteoarthritis, subchondral bone, microCT digital volume correlation, strain

## Abstract

Osteoarthritis (OA) is a chronic disease, affecting approximately one third of people over the age of 45. Whilst the etiology and pathogenesis of the disease are still not well understood, mechanics play an important role in both the initiation and progression of osteoarthritis. In this study, we demonstrate the application of stepwise compression, combined with microCT imaging and digital volume correlation (DVC) to measure and evaluate full-field strain distributions within osteoarthritic femoral heads under uniaxial compression. A comprehensive analysis showed that the microstructural features inherent in OA bone did not affect the level of uncertainties associated with the applied methods. The results illustrate the localization of strains at the loading surface as well as in areas of low bone volume fraction and subchondral cysts. Trabecular thickness and connectivity density were identified as the only microstructural parameters with any association to the magnitude of local strain measured at apparent yield strain or the volume of bone exceeding yield strain. This work demonstrates a novel approach to evaluating the mechanical properties of the whole human femoral head in case of severe OA.

## 1. Introduction

Osteoarthritis (OA) is a degenerative disease affecting synovial joints, encompassing the synovial membrane, synovial fluid, subchondral bone, cartilage, and the surrounding muscles and ligaments [1]. OA most commonly affects the middle age to elderly population, with approximately one third of people in the UK, over the age of 45 years (875 million people) seeking treatment for OA [2]. In the UK alone, approximately 97,000 primary hip replacements were performed in 2019, over 90% of which were for the indication of OA [3] illustrating the significant burden OA plays in society.

Whilst the etiology and pathogenesis of the disease are still not well understood, mechanics plays a role in the initiation, progression, and successful treatment of OA [4,5]. For example, uncorrected congenital hip dysplasia and femoral acetabular impingement, both of which alter the load transfer at the hip joint, have been shown to increase the risk of hip OA [6,7,8,9]. However, which specific mechanical parameters are most important and what their impact is on the disease process are still not well understood.

The mechanical properties of the bone tissue of femoral heads affected by OA are driven by their heterogeneous density and microarchitecture distribution and have been evaluated by means of mechanical testing of trabecular bone cylinders or cubes [10] extracted from different regions of the femoral head or by converting micro-computed tomography (microCT) images of those specimens into finite element models [11,12]. These studies suggest that microstructurally OA trabecular bone is more heterogenous, and tends to exhibit higher values of bone volume fraction (BV/TV) and trabecular thickness (Tb.Th), combined with decreased mineral fraction, when compared to healthy specimens. However, these studies have limited the analyses to portions of trabecular bone extracted from the main structure of the femoral head, neglecting the importance of the structural connectivity of the subchondral tissue and the effect of typical morphological features common in the OA such as osteophytes, bone cysts, and sclerotic bone [13]. Moreover, extracting specimens from the main structure requires the definition of a main testing direction, limiting the information about the mechanical competence of that particular portion of the tissue under different loading scenarios. 

The development of robust methods that allow comprehensive evaluation of the load distribution within the osteoarthritic bone structure will help to determine whether the altered bone microstructure affects the strain distribution in the tissue of the femoral head or if the remodeled structure maintains a homogeneous strain pattern. Digital volume correlation (DVC) is the only experimental technique that allows measurement of full-field three-dimensional (3D) displacement and strain within the structure. The procedure is based on 3D image datasets of the specimen acquired in unloaded and loaded configurations, which are then virtually registered to find the map of displacements representative of the deflection of the object under the applied load [14]. The displacements can then be differentiated into strains. Two extensive reviews of the applications of DVC for bone biomechanics have been published by Roberts et al. [15] and Grassi et al. [16]. This technique has been employed to evaluate full-field internal deformations in bone structures at the organ and tissue levels [17,18]. The DVC approach has also been used to study the effect of biomaterials and implants on the deformation of the bone tissue [19,20,21]. It has proved an auspicious approach for measuring internal strains of bone under various loading conditions and offers a crucial method for validation of finite element (FE) models of bone [22,23,24,25,26,27]. Nevertheless, the DVC approach has never been used to study the strain distribution in osteoarthritic femoral heads. The goal of this study was to develop a method to measure full-field strain distributions within osteoarthritic femoral heads. 

## 2. Materials and Methods

### 2.1. Specimen Preparation

Osteoarthritic femoral heads (*n* = 5) were obtained from patients undergoing routine total hip arthroplasty for clinically diagnosed OA (by X-ray) at the Royal Hallamshire Hospital (Sheffield). The specimens were stored in 70% ethanol at the time of retrieval for at least one month to reduce biological risk [28]. All research procedures were approved by the ethics committee of the South Yorkshire and North Derbyshire Musculoskeletal Biobank (Reference Number: STH 15691).

On the day prior to testing, specimens were removed from the ethanol solution, placed on absorbent paper for 30 min to remove excess liquid and potted in polymethylmethacrylate (PMMA, Technovit 4071, Kulzer GmbH, Hanau, Germany). The specimens were aligned such that the loading axis was perpendicular to the intraoperative surgical femoral neck cut. The femoral neck portion was potted in a 6 mm thick PMMA baseplate and a 6 mm top-plate was created that conformed to the superior surface of the femoral head (2–4 mm deep) to ensure an even application of the load. Parafilm was placed between the embedding material and the femoral head to ensure that the specimen did not adhere to the top plate whilst the resin set. After potting, specimens were placed in a custom-made loading device (Figure 1).

### 2.2. Test-jig

The jig was designed and manufactured to perform stepwise in situ mechanical loading within a microCT scanner (VivaCT80, Scanco Medical, Wangen-Brüttisellen, Switzerland). The jig comprised a radiolucent Perspex tube, 265 mm in length, which holds the loading platen. The load is applied by a loading screw (2 mm pitch) coupled to the loading platen via a thrust bearing that removes the torque, ensuring an axial load is applied to the loading platen. A miniature axial load cell (C9C 10 kN, HBM, Millbrook, UK) acquires the load on the specimen and a linear variable displacement transducer (LVDT WA T 20 mm, HBM) measures the on-axis displacement of the load platen.

### 2.3. Specimen Loading and µCT Imaging

All five specimens underwent stepwise loading with microCT imaging in situ. The images were acquired with a voxel size of 39 μm and the following scanner settings: 70 kVp, 114 mA, 300 ms integration time [29]. To capture the entire femoral head, image stacks ranged from 1006 slices (39.23 mm) to 1527 slices (59.56 mm), taking between 67 and 95 min per scan. A third-order polynomial, beam hardening correction algorithm, determined using a 1200 mg HA/cm^3^ wedge phantom, was applied to all scans as recommended by the manufacturer.

The loading regime consisted of the following: specimens were placed into the loading jig and 10 pre-loading cycles up to 0.3 kN were applied by manually rotating the loading screw using a torque wrench. The pre-load of 0.3 kN was sustained after the tenth cycle (Figure 2), and the specimen was left to relax for 15 min before the entire jig was placed into the microCT scanner. For each specimen, two repeated microCT images were acquired to evaluate the uncertainties of the DVC algorithm (Pre1 and Pre2). Following the repeated scans, each specimen underwent two more load steps up to 1.5 kN (LS1, Post1 image) and 3 kN (LS2, Post2 image), before a final load step attempting to achieve a post-yield load (LS3, Post3 image). The same relaxation time (15 min) was observed after the application of each load step and before scanning. A nominal pre-load of 0.3 kN was applied to minimize potential moving artifacts during the microCT scan. The load steps of 1.5 and 3.0 kN were selected based on preliminary trials on porcine femoral heads and represented loading levels within the elastic range and close to yield. Given that the porcine specimens were not sclerotic, it was expected that these limits would not exceed the yield limit of the OA femoral heads. This resulted in a total of five image stacks per specimen (Pre1, Pre2, Post1, Post2, Post3). The applied load and axial displacement of the loading platen were recorded at a frequency of 50 Hz; the sample rate was reduced to 1 Hz during the image acquisition (Spider 8 and Catman Easy software (version 4.2.1), HBM).

### 2.4. Image Processing and DVC Approach

All reconstructed images were converted to 8-bit (ImageJ) [30]. For each specimen the first image acquisition dataset (Pre1) was transformed by aligning the mean trabecular direction (MTD) parallel to the supero-inferior (SI) axis in the central coronal and sagittal slices; the fovea capitis femoris was then aligned perpendicular to the antero-posterior (AP) axis on the coronal and sagittal views as described in detail in Ryan et al. [13]. The Pre2 dataset was rigidly registered (Amira, v6.1) to the Pre1 image; Post1, Post 2 and Post3 data sets were then registered to Pre1 by applying the obtained transformation matrix to each of the subsequent datasets (Amira, v6.1, ThermoFisher Scientific, Ma, US). For the Pre1 scans, high frequency noise was removed by applying a Gaussian filter (support = 1 voxels, sigma = 0.5) followed by segmentation with a global threshold value (110 grey scale value, GSV) and a despeckle to remove any unconnected regions. The threshold was chosen based on the histogram as the best value that allowed discrimination between bone matrix and marrow.

To evaluate the microstructural parameters, the trabecular bone regions was extracted by the methods described in Ryan et al. [13]; briefly, a closing algorithm was used to create a mask of the whole bone, which was then used to determine the radius and centre of a best fit sphere within the femoral head. The slice including the centroid of the sphere was defined as the lowest slice of the region over which the analysis was performed and the cortical bone was removed by manual segmentation, drawing the contour every 25 slices and interpolating between slices. The following morphometric parameters were computed from the binarised images (CTAn, V1.17.7.2, Bruker, MA, USA): Trabecular Bone volume fraction (Tb.BV/TV) (%) was computed as the volume of binarised bone per unit volume of the segmented sample; Trabecular Thickness (Tb.Th) (µm) and Trabecular separation (Tb.Sp) (µm) were computed without model assumptions, using the three-dimensional local sphere-fitting method [31]; Trabecular Number (Tb.N) (mm^−1^) was calculated as the fractional volume divided by the thickness; and Connectivity Density (Conn.D) (mm^−3^), a measure of trabecular connectivity, is derived from the Euler number [32]. 

Considering that the used DVC software could read input images with maximum file size equal to 2 Gb, each image was resampled to 78 µm (rescaling factor of 0.5, bilinear interpolation, ImageJ) to ensure the entire bone volume could be analysed. For each specimen, the DVC analyses were performed on the resampled datasets. 

Details of the global DVC method used in this study (BoneDVC, https://bonedvc.insigneo.org/dvc/) are reported in previous studies in which the precision of the method has been evaluated for different bone structures [33,34,35]. Briefly, the approach overlaps a fixed grid onto the input images, with a set nodal spacing (NS). 3D deformable image registration (Sheffield Image Registration Toolkit, ShIRT) is then used to find the best set of displacements to be applied to each node of the grid that minimises the differences between the registered images. Displacements are linearly interpolated in between the nodes of the grid and optimal smoothing is applied to each registration. The DVC grid is then imported to a finite element (FE) software package (Ansys, V 15.0, Canonsburg, PA, USA) as a hexahedral mesh (78 µm isotropic resolution) and the computed nodal displacements are differentiated into strains by imposing the displacement field in each node as boundary condition. The six components of the strain tensor (εx, εy, εz, εxy, εyz, εxz), the principal compressive strain (εp3) and the Von Mises equivalent strain (εeqv) were computed in each node. An algorithm to remove all the nodes in the image outside a mask created for each femoral head was applied [36]. For the final DVC analyses, an optimal NS equal to 25 voxels (1950 µm) was applied after the uncertainties analyses described below.

### 2.5. Uncertainties Analyses of the DVC

To elucidate potential effects of morphometric variations within the osteoarthritic femoral heads (e.g., in regions of high bone volume fraction (BV/TV) or due to the presence of cysts) on the uncertainties of the DVC measurements, a single specimen was selected to undergo detailed analysis at the original image resolution (39 µm). Four volumes of interest (VOIs) were defined in the following way: VOI0 incorporated the largest volume inscribed within the femoral head (9999 × 9999 × 727 voxels), VOI1, VOI2, VOI3 were of equal size (600 × 600 × 785 voxels) and represented regions containing osteophytes, central trabeculae and subchondral cysts, respectively. DVC analyses were performed on each VOI under the zero-strain condition (repeated scan), and by virtually deforming the Pre2 images to the equivalent of 1% and 5% apparent strain (repeated virtually deformed condition) [37]. All analyses were performed with nodal spacings of 25, 50, 75 and 100 voxels (equivalent to 975, 1950, 2925 and 3900 µm, respectively). 

To determine an optimal nodal spacing and to evaluate the precision of the DVC approach, uncertainty analyses were performed for each specimen with the zero-strain condition, on the resampled repeated images [38,39]. NSs equal to 13, 25, 35 and 50 voxels (equivalent to 1014, 1950, 2730 and 3900 µm, respectively) were evaluated. These NSs were chosen as approximately equivalent to the NSs used in the DVC analysis on the original resolution images. To evaluate potential effects of resampling, uncertainty measures for the resampled and original resolution were also compared for one specimen in the zero-strain condition.

For all analyses, the effects of image noise outside the border of the specimens were removed by performing the registration only within a mask of the external contour of the bone. For each NS, and in each VOI and specimen, the systematic and random errors were computed as the average and standard deviation for each component of strain and displacement, respectively [39]. The mean absolute error (MAER) and standard deviation of the error (SDER) were computed as average and standard deviation of the average of the absolute value of the components of strain in each node [38].

### 2.6. DVC Analyses and Evaluation of Strain

For each specimen, DVC analyses were performed for the three load steps (LS1: Pre1 vs Post1; LS2: Pre1 vs Post2; LS3: Pre1 vs Post3). For each load step, the maximum, minimum, and median values of the axial and shear strain components, principal compressive strain and Von Mises equivalent strain were computed. For LS2 and LS3, the volume of bone exhibiting compressive strains exceeding 10,000 µε, typical values representative of bone yield [40], was also computed.

DVC strain outputs were mapped onto 3D renderings of the bone specimens, to allow qualitative inspection of local morphometric features that may influence the strain distributions. Two-tailed Spearman’s rank correlation (r_s_) was used to explore any associations between apparent morphometric parameters and the strain measurements at LS2 and at LS3.

## 3. Results

The specimens were retrieved from patients with a mean age of 65.4 years (standard deviation (SD) = 9.7, range = 52–82). Consistent with microarchitectural changes observed in OA, the specimens exhibited sclerotic bone, with increased trabecular thickness in the subchondral region, in combination with significant voids or cysts. The morphometric measurements for each specimen are reported in Table 1. A detailed comparison of the morphometric properties for these OA specimens with healthy specimens can be found in Ryan et al. [13].

### 3.1. Mechanical Loading

All specimens successfully underwent stepwise loading by three load steps after the repeated scans. The load-displacement curves for all specimens are shown in Figure 2. The maximum applied load achieved across all five specimens was between 5.37 and 6.26 kN.

### 3.2. Uncertainties Analyses of the DVC Approach

No regional effects across the four different VOIs were observed for DVC uncertainties, which were consistent for all the zero-strain and the repeated virtually deformed cases (Figure 3). For the repeated virtually deformed images, the systematic and random errors were highest along the loading axis for each VOI (for applied εzz equal to 1%: 988 µε; for applied εzz equal to 5%: 951 µε). The error in this component could be reduced by removing the top and bottom slices perpendicular to the z direction, highlighting errors concentrating in the border that are expected from the used global DVC approach whilst the error associated with the other strain components were relatively independent from the number of slices removed. Consequently, for the repeated virtually deformed images, the top and bottom 2 slices (3.9 mm) were not included in the DVC measurements to minimise errors on the surfaces associated with the deformed shape. The MAER and SDER across all four VOIs, with a nodal spacing of 50 voxels (1.950 mm) and for all loading conditions, are reported in Table 2. Figure 3 shows the outputs for compressive strain observed across all four VOIs under the zero load and 1% virtual compression load.

Resampling the images reduced slightly the errors for the zero-strain analyses in the whole femoral head (MAER from 599 to 258 µε, and SDER from 485 to 311 µε). As expected, across all specimens, uncertainties decreased with increasing NS, following a logarithmic pattern [34]. At the resampled resolution of 78 µm, a NS of 25 voxels (1.950 mm) yielded the best compromise between spatial resolution and accuracy. Across all 5 specimens, a NS of 25 voxels yielded a MAER between 399 and 729 µε and an SDER between 437 and 612 µε.

### 3.3. Digital Volume Correlation (DVC)

At LS2 (3 kN applied load) the median principle compressive strain (εp3) measured was 2026 µε (range across the specimens: 1695–4237 µε) and on average 18% (range: 4–34%) of the bone volume exhibited εp3 greater than 10,000 µε. At the highest load step (~6 kN applied load) median εp3 increased to 4489 µε (range across the specimens: 3401–14,836 µε) and the average volume of bone exhibiting strains above 10,000 µε increased to 33% (range 22–61%). Histograms illustrating the distribution of the principal compressive strain (εp3) and Von Mises equivalent strain (εeqv) at each load step and for each specimen are shown in Figure 4. Figure 5 and Figure 6 and Appendix A show the similar trends observed across all strain components for all specimens with an increasing percentage of bone volume exhibiting strains above 10,000 µε with increased loading from LS2 to LS3.

DVC outputs for the three shear strain components (εxy, εyz, εxz) and the principle compressive (εp3) for Specimen#1 and Specimen#4 are shown in Figure 5 and Figure 6, respectively.

Similar figures for the remaining specimens are available in Appendix A. The visualisations illustrate the localisation of strains at the fixed surface and at the surface of the applied load, diffusing inwards towards the center of the bone at LS2. At LS3, the strain increases in magnitude and the volume of bone at exceeding yield strain, but additionally, regions of high shear strain appear around areas corresponding to low BV/TV. The localisation of strain at the loading and fixed surface in LS2 was similar across all specimens, with localisation of large shear strains around cystic or low BV/TV regions at LS3. Histograms for all strain components across the three load steps (LS1, LS2 and LS3) are shown in Figure 5 for Specimen#1 and in Figure 6 for Specimen#4 (similar plots for the other specimens are provided in Appendix A). 

Results for the relationship between morphometric and strain outputs are reported in Table 3. A moderate linear degree of association was observed between the Conn.D and the median principal compressive strain (med-εp3) computed at LS3 (r_s_ = −0.90, *p* = 0.037). Significant correlations were observed between Tb.Th and med-εp3 at LS2 (r_s_ = 0.90, *p* = 0.037), and at LS3 (r_s_ = 1.00, *p* <0.01). The volume of bone exceeding 10,000 µε at LS3 had strong linear associations with both Tb.Th (r_s_ = −0.90, *p* = 0.037) and Conn.D (r_s_ =1.00, *p* < 0.01). No other statistically significant relationships were observed.

## 4. Discussion

The goal of this study was to evaluate full-field strain distributions within the whole OA femoral head under compressive loads. The results from this study provide, for the first time, an insight into the distribution of strains with respect to different features of the complex trabecular structure in OA femoral head bone specimens.

A comprehensive evaluation of the measurement uncertainties determined that regional variations in morphometry commonly seen in OA bone microarchitecture (e.g. cysts, sclerosis) did not influence the accuracy of the DVC approach. Four VOIs were evaluated and the SDER remained below 644 µε for virtual deformations up to 1% and NS of 1.95 mm. When the repeated image stacks underwent virtual compressive deformation, the most superior and inferior slices were replaced with black voxels, which introduced sharp gradients of grey-levels and artifacts in the assessment of the DVC precision, similar to that observed in a previous study on synchrotron images of cortical bone [37]. Removing slices at the superior and inferior surfaces reduced this error, whilst all other component strains were unaffected. Resampling did not increase the errors of the DVC and for the optimal NS used in the study (1.95 mm) all demonstrated uncertainties of a few hundred microstrain, are at least one order of magnitude below the yield strain for bone tissue (10,000 µε), allowing discrimination of the regions beyond yield. Nevertheless, to evaluate more precisely the local strains in single trabeculae or within the cortical bone, input images with higher resolution should be used. It should be noted that higher errors were localised in the distal surface of the bone, likely attributed to border effects and that were less pertinent for the analyses performed in this study.

The DVC approach allowed visualisation and quantification of the strains within the bone volumes under stepwise compressive load. The 3D renderings (Figure 4) showed that regions of high strain primarily localised at the base of the specimen (where fixation occurred) and at the loading surface, whilst the volume of bone within the central region remained relatively unloaded. In particular, it should be noted that the higher strains seem to localise around the heterogeneous structure of the superior femoral head. Previous studies have shown that microstructural changes within the OA femoral head lead to sclerosis and subsequent increased BV/TV in the subchondral bone with a coincident reduction in BV/TV in regions medial and lateral to the principle loading axis [13]. 

In excised trabecular bone specimens, apparent level mechanical properties are largely influenced by the underlying microstructural parameters [10,27,41,42,43,44,45,46,47]. Whilst previous work has identified BV/TV as a strong predictor of stiffness and strength in excised specimens, within our cohort we did not find any statistically significant relationships between BV/TV and any of the strain parameters evaluated. Within the whole femoral head, linear associations were only identified between the strain and Tb.Th or Conn.D at LS3. Tb.Th is a measure of the average thickness of the trabecular structure. With increasing Tb.Th, a smaller volume of bone exceeded the yield threshold (10,000 µε), suggesting the trabecular thickening (consistent with sclerotic changes that occur in OA) is able to sustain higher apparent loads by efficient distribution of the load throughout the entire bone network. The strongest associations with local strains were observed for Conn.D, which is a descriptor of the degree of connectivity of the underlying architecture. In this cohort of specimens, a higher measure of Conn.D (i.e., increased connectivity) resulted in higher values of median εp3 and an increased volume of bone exceeding yield strain, however this was also associated with a decrease in Tb.Th. Together these findings suggest that the sclerotic bone (i.e., more plate like structure) expresses as a reduction in connectivity resulting in a reduction in magnitude of strain and volume of high strain bone. It is important to note, however that the morphometric measures reported herein were computed across the entire trabecular bone region in the femoral head and represent a global measure. The illustrations shown in Figure 5 and Appendix A, however, demonstrate the large heterogeneity in architecture throughout the entire femoral head. In several specimens, the increased BV/TV and sclerotic changes in the subchondral region is associated with a decrease in BV/TV in regions medial and lateral to the mean trabecular axis [13]. The DVC outputs demonstrate that at high apparent loads, higher magnitude strains and localisations tend to occur specifically around cystic regions, or regions with lower BV/TV. Those specimens with a more homogenous distribution of bone exhibited less heterogeneity in the strain patterns at LS3. 

Some regions of bone exhibited strains above 10,000 µε even under 1.5 kN load. These high strain regions were limited to the base of the specimen, the same regions that displayed high systematic and random errors in the analysis of measures of uncertainty, and as such, the magnitude of these strains should be interpreted with caution.

Some of the limitations to note in the study: Firstly, the sample size is small, and prospective relationships and trends between mechanical and morphometric measures should be interpreted with this caveat. While the low sample size does not allow clinical inferences from the findings, this feasibility study has highlighted the potential of the approach that will be used in the future to analyze a larger cohort of specimens extracted from patients with different levels of OA. Moreover, a control group of femoral heads extracted from healthy donors (cadaver study) would be needed to identify the effect of disease progression on the strain distribution. Additionally, the specimens were stored in 70% ethanol before testing. While this storage approach has been found to have minimal effects on the bone elastic properties [48], little is known about its effect on post-yield properties and strain distribution. Therefore, in future studies that aim to provide a clinical interpretation of the biomechanical results, fresh frozen tissues should be used. Comparison between specimens was performed under 3 kN load, where the apparent load was high enough to elicit meaningful local strain measures and the load was consistent across specimens. Apparent stress or strain was not computed and variances in bone size and volumes means the resultant apparent stress or strain applied to each specimen is likely different among the cohort. However, this highlights the importance of computation models, which are better able to address these issues. In addition, the load screw mechanism for the application of high loads was dependent on user strength to reach failure loads. Consequently, whilst we endeavored to apply loads equivalent to yield stress, this was not achieved for all cases. Finally, it should be noted that the applied load was not representative of physiological loading but was applied relative to the surgical cut face of the specimen. The graphical renderings illustrate, however, that whilst loading was not applied along the mean trabecular axis, the load vector relative to the mean trabecular axis appears consistent among specimens. 

## 5. Conclusions

In conclusion, this work has demonstrated the use of stepwise mechanical loading combined with microCT imaging to visualise the heterogeneous deformation of the microstructure within the OA femoral heads under compression. The DVC approach enabled identification of regions of high strain localisation and allowed qualitative analysis of the localisation of strains relative to bone microarchitecture. Further work is needed to evaluate the microstructural parameters specifically in the regions of high strain, and to increase the sample size to identify trends in strain distributions and localisations. Moreover, after proper validation with DVC data, micro-FE analyses will be used to evaluate the strain patterns under a variety of loading regimes. 

## Figures and Tables

**Figure 1 materials-13-04619-f001:**
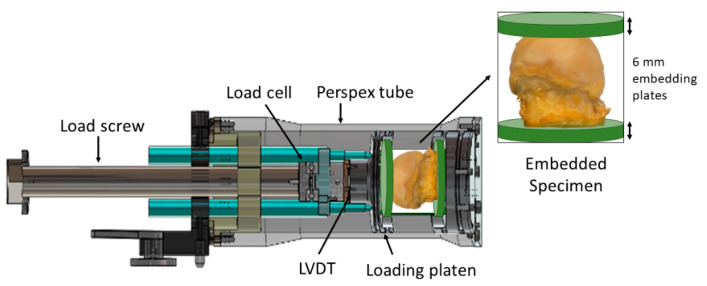
Specimens were embedded in 6mm thick acrylic resin plates, with the top plate created such that it conformed to the rounded surface of the femoral head. The embedded specimens were placed in the mechanical test jig for loading. The loading jig comprised a load screw to apply the compression through a loading platen, a load cell and linear variable displacement transducer (LVDT) to measure applied load in displacement of the loading platen, respectively. The system was encased in a Perspex tube so that it remained visible throughout testing.

**Figure 2 materials-13-04619-f002:**
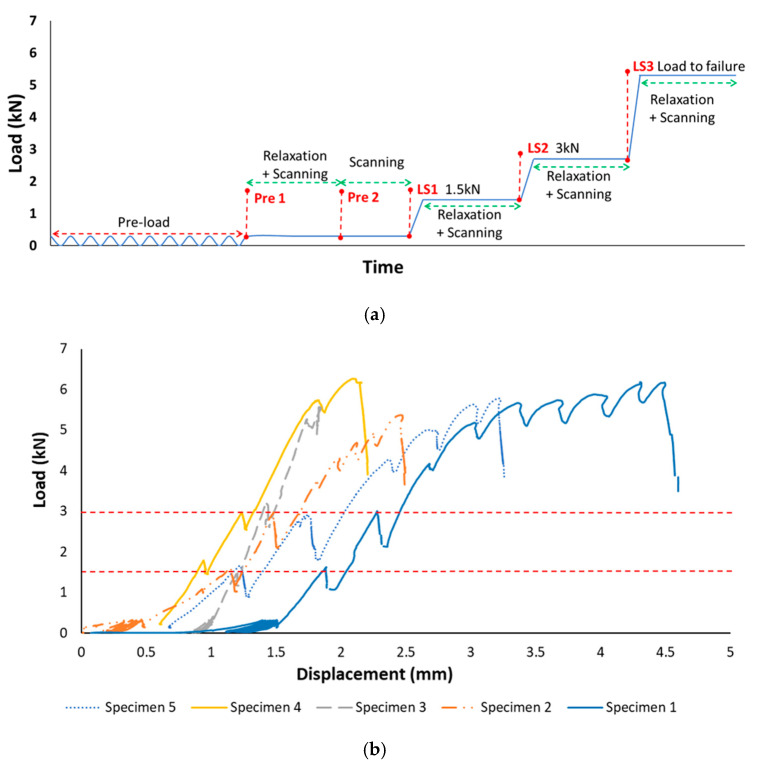
(**a**) The loading regime applied to each specimen consisted of 10 cycles up to 0.3 kN load, and two image acquisitions (Pre1 and Pre2); a 1.5 kN load was applied and a “post load” scan was performed (Post1); the load was increased to 3 kN and another scan acquired (Post2); finally the load was applied until it appeared that apparent failure had been achieved and a final scan was performed (Post3). A 15 min. relaxation period was observed following the application of any load and prior to scanning; (**b**) Load-Displacement curves recorded for the 5 specimens. After the load step at 3 kN load, the “saw-tooth” profile is a result of attempting to apply the failure load via the load screw.

**Figure 3 materials-13-04619-f003:**
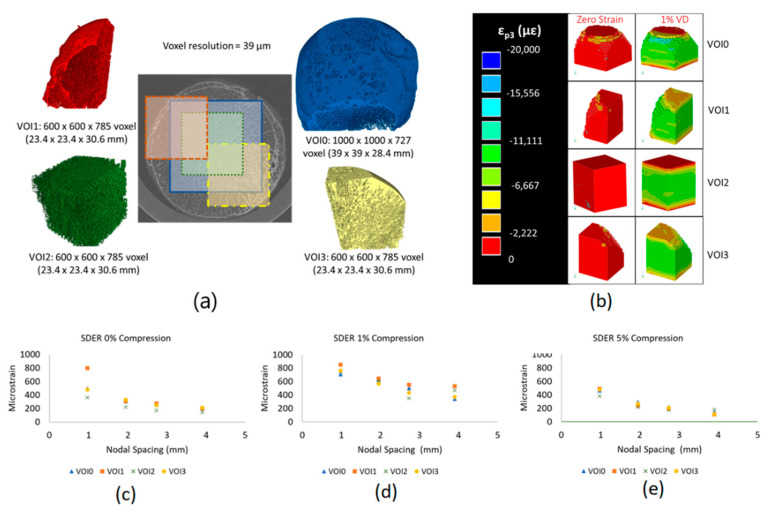
(**a**) Volumes of interest (VOIs) were extracted from one of the OA specimens to evaluate uncertainties within a representative total volume (blue), central region in the trabecular bone (green) and regions containing cysts (red) and sclerotic bone (yellow). (**b**) DVC strain fields for the four VOI’s under zero-strain (repeated images) and 1% virtual compressive apparent strain (1% VD). (**c**–**e**) Little difference in the measures of standard deviation of the error (SDER) measures was observed across each of the four VOI’s, and with nodal spacings equivalent to 1.01, 1.95, 2.73 and 3.90 mm under each of the 0% compression and 1% and 5% virtual compressive apparent strain cases.

**Figure 4 materials-13-04619-f004:**
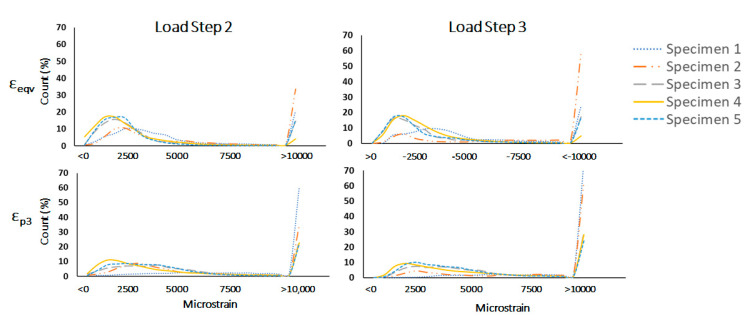
Frequency distributions of strain values output by the DVC for Von Mises equivalent strain (εeqv) (**TOP**) and principle compressive strain (εp3) (**BOTTOM**) at LS2 (**LEFT**) and LS3 (**RIGHT**). At LS3, there is a distinct increase in the volume of tissue exhibiting εp3 > 10,000 µε, with a concurrent reduction in tissue volumes exhibiting low strain values, which was consistent across all specimens.

**Figure 5 materials-13-04619-f005:**
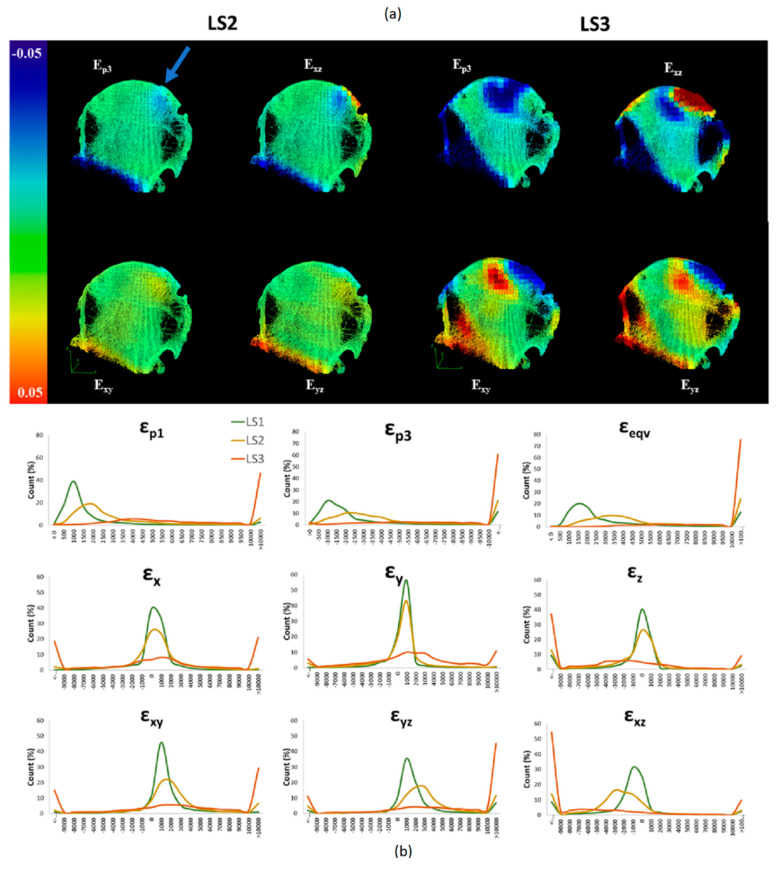
(**a**) 3D Renderings of 5 mm thick slices of Specimen 1 overlaid with the DVC output strain maps for (εp3, εxy, εyz and εxz). The images qualitatively illustrate the localisation of strain at LS2 (~3 kN) (LEFT) and increase and propagation of strains at LS3 (~ 6 kN) (RIGHT). The load was applied on the superior side of the femoral head (blue arrow) perpendicular to the cut surface of the femoral neck. (**b**) Histograms illustrating frequency distribution for strain components (εp1 εp3 εeqv εx εy εz εxy εyz εxz) and incremental load steps (LS1 (green), LS2 (yellow) and LS3 (orange). Large volumes of bone exceeding the yield strain of bone (10,000 µε) are present at the apparent yield load step (LS3).

**Figure 6 materials-13-04619-f006:**
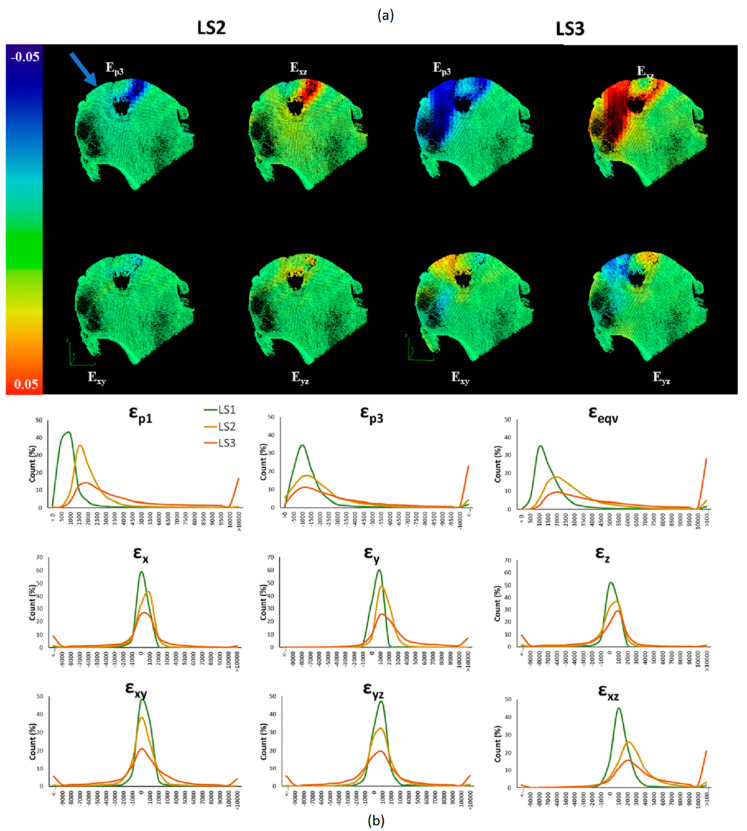
(**a**) 3D Renderings of 5 mm thick slices of Specimen 4 overlaid with the DVC output strain maps for (εp3, εxy, εyz and εxz). The images highlight the localisation of strains around the large cyst at LS2 (LEFT), and LS3 (RIGHT). The load was applied on the superior side of the femoral head (blue arrow) perpendicular to the cut surface of the femoral neck. (**b**) Histograms illustrating frequency distribution for strain components (εp1 εp3 εeqv εx εy εz εxy εyz εxz) and incremental load steps (LS1 (green), LS2 (yellow) and LS3 (orange). Large volumes of bone exceeding the yield strain of bone (10,000 µε) are present at the apparent yield load step (LS3).

**Table 1 materials-13-04619-t001:** Morphometric measurements for all specimens (*n* = 5).

Specimen	Age (Years)	Gender	Tb.BV/TV (%)	Tb.Th (µm)	Tb.Sp (µm)	Tb.N (mm^−^^1^)	Conn.D (mm^−3^)
Specimen #1	62	F	17.7	243	1297	0.728	8.26
Specimen #2	64	F	24.6	250	743	0.983	6.41
Specimen #3	52	F	25.7	271	801	0.949	5.23
Specimen #4	82	M	21.3	269	1075	0.793	5.06
Specimen #5	67	F	22.5	304	1098	0.740	3.20

**Table 2 materials-13-04619-t002:** Mean Absolute Error (MAER) and Standard Deviation of the Error (SDER) across all VOIs for the 3 different loading conditions considered for a NS equal to 50 voxels (1.95 mm).

	MAER (µε) Range VOI0–VOI3				SDER (µε) Range VOI0–VOI3			
Load case	VOI0	VOI1	VOI2	VOI3	VOI0	VOI1	VOI2	VOI3
Zero load	529	568	376	495	306	310	221	331
1% virtual compression	469	488	527	460	296	244	215	270
5% virtual compression	611	530	486	511	630	644	611	569

**Table 3 materials-13-04619-t003:** Spearman’s rank correlation coefficient for exploring linear associations between strain outputs and morphometric properties (*n* = 5).

		Morphometric Parameter				
Load Step	Strain Parameter	Tb.BV/TV (%)	Tb.Th (µm)	Tb.Sp (µm)	Tb.N (mm^−1^)	Conn.D (mm^−3^)
Load Step 2	med-ε_p3_	NS *p* = 0.75	r_s_ = 0.90 *p* = 0.037	NS *p* = 0.624	NS *p* = 0.624	NS *p* = 0.104
	Percent > 10,000 µε	NS *p* = 0.87	NS *p* = 0.322	NS *p* = 0.741	NS *p* = 0.741	NS *p* = 0.172
Load Step 3	med-ε_p3_	NS *p* = 0.39	r_s_ = 1.0 *p* < 0.01	NS *p* = 0.873	NS *p* = 0.873	r_s_ = −0.90 *p* = 0.037
	Percent > 10,000 µε	NS *p* = 0.747	r_s_ = −0.90 *p* = 0.037	NS *p* = 1.0	NS *p* = 1.0	r_s_ = 1.0 *p* < 0.01

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
