# Peer review of "Heterogeneous Strain Distribution in the Subchondral Bone of Human Osteoarthritic Femoral Heads, Measured with Digital Volume Correlation"

_materials, 2020, doi:10.3390/ma13204619_

Round 1

Reviewer 1 Report

Thank you for giving me this opportunity to review the research article entitled, "Heterogeneous strain distribution in the subchondral bone of human osteoarthritic femoral heads, measured with digital volume correlation"

I here carefully reviewed the submitted set of the manuscript and found it merits of publication.

In general, the study design is appropriate, the methods employed is reasonable and the discussions are easy to follow.

Just let me ask you to add the clinical feedbacks for the patients with OA in femoral head considering and making much use of the results obtained in the present study should be well discussed in the discussion section. 

Author Response

Considering that this feasibility study has investigated the outputs of only 5 specimens, we believe that we can’t provide any clinical feedback for the patients. Nevertheless, in future work on a larger cohort of specimens extracted from patients with different levels of OA we will be able to provide a clinical feedback.

We have added a sentence to the discussion to highlight this.

Reviewer 2 Report

The manuscript is well written and is original.  I recommend for publication. However, I have minor comments shown below,

The strain calculation is not clearly written, needs further elaboration?

The details of Mesh, such as element size element number employed in Ansys were missed.

Author Response

We thank the reviewers for their comments.

The information requested by the reviewer has been added into the manuscript.

Reviewer 3 Report

The authors present an interesting study about the use of digital volume correlation for measuring the mechanical behavior of the femoral heads of five patients with osteoarthritis. Despite the rather low number of samples, the study provides valuable insights about the feasibility of this method in assessing strain distributions within this type of tissue and as such a good preliminary basis towards future exploration of this methodology.

Abstract

- “A comprehensive analysis showed the”: please change to ‘A comprehensive analysis showed that the”

Materials and Methods

- Please mention briefly how OA was clinically diagnosed.

- Can the authors briefly mention how this specific loading regime was selected/optimised?

Author Response

We thank the reviewers for their comments. We have addressed the comments as follows: 

Abstract

“A comprehensive analysis showed the”: please change to ‘A comprehensive analysis showed that the”

 This has been amended

Materials and Methods

Please mention briefly how OA was clinically diagnosed.

We have added in the manuscript that the OA was diagnosed with X-Rays.

Can the authors briefly mention how this specific loading regime was selected/optimised?

The following has been added to Section 2.3 of the Methods:
A nominal pre-load of 0.3kN was applied to minimise potential moving artifacts during the microCT scan. The load steps of 1.5kN and 3.0kN were selected based on preliminary trials on porcine femoral heads and represented loading levels within the elastic range and close to yield. Given that the porcine specimens were not sclerotic, it was expected that these limits would not exceed the yield limit of the OA femoral heads.

Reviewer 4 Report

Thanks for submitting this manuscript, which presents the Heterogeneous strain distribution in the subchondral bone of human osteoarthritic femoral heads.

I have carefully read your manuscript with great interest.

I think that it should sound very interesting for readers and this paper overall well written.

This study is well designed and conducted.

I have a few minor comments.

Abstract

It need to clear the meaning of abbreviations as “BV/TV”.

Material & Method

The following morphometric parameters: Trabecular Bone volume fraction (Tb.BV/TV) (%), Trabecular Thickness (Tb.Th) (μm), Trabecular separation (Tb.Sp) (μm), Trabecular Number (Tb.N) (mm-1) and Connectivity Density (Conn.D) (mm-3).

How to definite the morphometric parameters? There is no explanation. It is uncomfortable for reader.

Author need to add the definition of morphometric parameters using simple figure of trabecular bone structure.

Author Response

Abstract

It need to clear the meaning of abbreviations as “BV/TV”.

This has now been amended to read bone volume fraction in the abstract.

Material & Method

The following morphometric parameters: Trabecular Bone volume fraction (Tb.BV/TV) (%), Trabecular Thickness (Tb.Th) (μm), Trabecular separation (Tb.Sp) (μm), Trabecular Number (Tb.N) (mm-1) and Connectivity Density (Conn.D) (mm-3).

How to definite the morphometric parameters? There is no explanation. It is uncomfortable for reader.

The paragraph of the methods where these parameters are introduced has been amended to clearly define the parameters and how these were determined.

Author need to add the definition of morphometric parameters using simple figure of trabecular bone structure.

The definition of the morphometric has now been defined in more detailed, and we feel this should suffice explanation of the structure. Since the morphometry is not the main focus, we have not included a figure for this.

Reviewer 5 Report

Abstract and lines 43: do not use BV/TV without previously describing the abbreviation meaning

Line 33: please add a reference to the sentence

Lines 68-69: I suggest moving this idea to the discussion section

Lines 72-73: mean age, range… are study results. Please move this information to the results section

Section 2.1: the specimens were stored in 70% ethanol for at least one month. The authors do not describe a rehydration step, so we can assume they performed the tests on dehydrated samples, presenting different mechanical properties, which compromise the obtained results. Is this correct? If so, this must be stated and discussed.

Figure 1 caption needs to be changed. It repeats information already presented and does not appropriately describe the image

Figures 2 and 3 (graphs) quality is low and needs to be improved

Lines 309-311: the authors stated that “A comprehensive evaluation of the measurement uncertainties determined that regional variations in morphometry commonly seen in OA bone microarchitecture (e.g. cysts, sclerosis) did not influence the accuracy of the DVC approach”. Although it is expected the used samples present the referred variations, that is nowhere described in the manuscript to support this information. A brief description of the images reporting this information should be added.

Please put the “p” referent to p-value in italic form

In my opinion, control samples of normal bone are missing. The heterogeneous deformation observed can also be presented in normal samples and not be derived from the OA bone. Does this methodology has already been applied to normal samples? Please present the data of normal bone. This is fundamental to validate the technique.

Please put the references in the appropriate journal style

Author Response

Abstract and lines 43: do not use BV/TV without previously describing the abbreviation meaning

This has now been amended to read bone volume fraction in the abstract.

Line 33: please add a reference to the sentence

The following references have been added:

Felson, D.T., Osteoarthritis as a disease of mechanics. Osteoarthritis and cartilage, 2013. 21(1): p. 10 15.

Varady, N.H. and A.J. Grodzinsky, Osteoarthritis year in review 2015: mechanics. Osteoarthritis and cartilage, 2016. 24(1): p. 27-35

Lines 68-69: I suggest moving this idea to the discussion section

This has been moved from the last sentence of the introduction to the first paragraph of the discussion and amended as follows:
The results from this study provide, for the first time, an insight into the distribution of strains with respect to different features of the complex trabecular structure in OA femoral head bone specimens.

Lines 72-73: mean age, range… are study results. Please move this information to the results section

This has been moved to the first sentence of the results section.

Section 2.1: the specimens were stored in 70% ethanol for at least one month. The authors do not describe a rehydration step, so we can assume they performed the tests on dehydrated samples, presenting different mechanical properties, which compromise the obtained results. Is this correct? If so, this must be stated and discussed.

Considering the size of the specimens it would be difficult to rehydrate homogenously the internal portion of the femoral head, even using vacuum procedures. Moreover, it would be difficult to control the hydration in such long in situ mechanical testing. Therefore, we have decided to perform the tests in dehydrated conditions. Considering the relatively long time in ethanol to reduce the risk of biological contamination we assume that a similar status of dehydration has been achieved for the five considered specimens. The storage in Ethanol 70% has been shown not to affect the elastic properties of the bone tissue (Linde&Sorensen) but the reviewer is correct in stating that some of the postyield properties may be affected by the storage in ethanol.

This was a feasibility study to highlight the potential of the combined in situ testing, microCT imaging and DVC analyses to measure the heterogeneous deformation within the complex structure of the OA femoral head. The next step is to apply the procedure to a large cohort of fresh frozen specimens from patients with different degrees of OA.

A limitation has been added in the discussion.

Figure 1 caption needs to be changed. It repeats information already presented and does not appropriately describe the image

Figure 1 caption has now been amended

Figures 2 and 3 (graphs) quality is low and needs to be improved

The figure quality has been improved

Lines 309-311: the authors stated that “A comprehensive evaluation of the measurement uncertainties determined that regional variations in morphometry commonly seen in OA bone microarchitecture (e.g. cysts, sclerosis) did not influence the accuracy of the DVC approach”. Although it is expected the used samples present the referred variations, that is nowhere described in the manuscript to support this information. A brief description of the images reporting this information should be added.

The first paragraph of the results has been amended to read as follows:
Consistent with microarchitectural changes observed in OA, the specimens exhibited sclerotic bone, with increased trabecular thickness in the subchondral region, in combination with significant voids or cysts. The morphometric measurements for each specimen are reported in Table 1. A detailed comparison of the morphometric properties for these OA specimens with healthy specimens can be found at (Ryan et al, 2020) [13]

Please put the “p” referent to p-value in italic form

All ‘p’s’ with reference to p-values have been changed to italic

All ‘p’s’ with reference to p-values have been changed to italic

In my opinion, control samples of normal bone are missing. The heterogeneous deformation observed can also be presented in normal samples and not be derived from the OA bone. Does this methodology has already been applied to normal samples? Please present the data of normal bone. This is fundamental to validate the technique.

We agree with the reviewer that a control group would be needed to clarify the effect of OA on the heterogeneity of the deformation in the femoral head. Nevertheless, this feasibility study aimed at showing the potential of the used approach in identifying heterogeneity in the strain distribution within the femoral head and, therefore, the lack of a control group does not affect the conclusion of the study.

Of course, if in future studies we will use this approach to identify effects of the disease (OA) we would need a control group from healthy subjects. A limitation has been added in the discussion to clarify this point.

Please put the references in the appropriate journal style

The appropriate reference style has been downloaded and applied to the manuscript

Round 2

Reviewer 5 Report

The manuscript has been significantly improved.

The references still need correction, as previously pointed to the authors.